# Salinity Stress Ameliorates Pigments, Minerals, Polyphenolic Profiles, and Antiradical Capacity in Lalshak

**DOI:** 10.3390/antiox12010173

**Published:** 2023-01-11

**Authors:** Umakanta Sarker, Md. Nazmul Hossain, Shinya Oba, Sezai Ercisli, Romina Alina Marc, Kirill S. Golokhvast

**Affiliations:** 1Department of Genetics and Plant Breeding, Faculty of Agriculture, Bangabandhu Sheikh Mujibur Rahman Agricultural University, Gazipur 1706, Bangladesh; 2Laboratory of Field Science, Faculty of Applied Biological Sciences, Gifu University, Yanagido 1-1, Gifu 501-1193, Japan; 3Department of Horticulture, Faculty of Agriculture, Ataturk University, 25240 Erzurum, Turkey; 4Food Engineering Department, Faculty of Food Science and Technology, University of Agricultural Sciences and Veterinary Medicine, 400372 Cluj-Napoca, Romania; 5Siberian Federal Scientific Center of Agrobiotechnology RAS, 2b Centralnaya Street, 630501 Krasnoobsk, Russia

**Keywords:** color attributes and pigment, minerals, beta-carotene, ascorbic acid, HPLC–UV, DPPH, ABTS^+^, polyphenolic profiles, antiradical capacity, salinity stress

## Abstract

Previous studies have shown that salinity eustress enhances the nutritional and bioactive compounds and antiradical capacity (ARC) of vegetables and increases the food values for nourishing human diets. Amaranth is a salinity-resistant, rapidly grown C_4_ leafy vegetable with diverse variability and usage. It has a high possibility to enhance nutritional and bioactive compounds and ARC by the application of salinity eustress. Hence, the present study aimed to evaluate the effects of sodium chloride stress response in a selected Lalshak (*A. gangeticus*) genotype on minerals, ascorbic acid (AsA), Folin–Ciocalteu reducing capacity, beta-carotene (BC), total flavonoids (TF), pigments, polyphenolic profiles, and ARC. A high-yield, high-ARC genotype (LS6) was grown under conditions of 0, 25, 50, and 100 mM sodium chloride in four replicates following a block design with complete randomization. We recognized nine copious polyphenolic compounds in this accession for the first time. Minerals, Folin–Ciocalteu reducing capacity, AsA, BC, pigments, polyphenolic profiles, and ARC of Lalshak were augmented progressively in the order: 0 < 25 < 50 < 100 mM sodium chloride. At 50 mM and 100 mM salt concentrations, minerals, AsA, Folin–Ciocalteu reducing capacity, BC, TF, pigments, polyphenolic profiles, and ARC of Lalshak were much greater than those of the control. Lalshak could be used as valuable food for human diets as a potent antioxidant. Sodium chloride-enriched Lalshak provided outstanding quality to the final product in terms of minerals, AsA, Folin–Ciocalteu reducing capacity, BC, TF, pigments, polyphenolic profiles, and ARC. We can cultivate it as a promising alternative crop in salinity-prone areas of the world.

## 1. Introduction

Consumers’ acceptability largely depends on the color, flavor, and taste of products. Presently, coloring food products has attracted the favor of consumers. These products have received much interest from consumers in aesthetic, nutritional, and safety aspects of food. The demand for natural pigments such as betacyanins, betaxanthins, betalains, anthocyanin, amaranthine, carotenoids, and chlorophylls is increasing considerably day by day. The selected Lalshak genotype is bright red–violet due to the presence of abundant betalains. Leaves of amaranths are an exclusive source of betalains (betaxanthins and betacyanins) that have a strong antiradical capacity (ARC) [1]. Betalains could be used as a food colorant in low-acid foods as they have higher pH stability than anthocyanins [2]. Amaranthine, a major pigment of betacyanins in Lalshak, has very strong ARC. It could be used as a substitute for betanins from red beets in food colorants and natural antioxidants [1]. Betacyanins, betaxanthins, and carotenoids are also free radical scavengers (antioxidants) [3] that play an important role in human health [4]. The active ingredients of betalains and carotenoids provide anti-inflammatory properties to our food and reduce the risk of cardiovascular disease and lung and skin cancers [4,5,6]. The natural properties of betalains and carotenoids of the amaranth genotype enable these compounds to be used as additives for drugs, food, and cosmetic products [7].

Amaranth is an excellent source of minerals and a unique source of pigments—such as betalains, chlorophylls, carotenoids [8,9,10] with strong ARC [11,12], polyphenolic profiles, BC, and ascorbic acid (AsA) [13,14] with high ARC [15,16]—that has remarkable contributions to the food industry as these compounds scavenge reactive oxygen species (ROS) [17,18,19] in the human body. It may be utilized in various medications, especially neuroprotective [20], antimicrobial [21,22,23], antiviral [24], antiulcer [25], anthelmintic [26,27,28,29], hepatoprotective [30,31,32,33], anticancer [34,35], anti-inflammatory [36,37], anti-hyperlipidemic [38,39,40], antimalarial, antidiabetic, and snake antidote [41,42,43] medicine. The concentration of secondary plant metabolites was enhanced with a degree of salinity that ultimately accelerated plant defense mechanisms against oxidative stress [44]. Sodium chloride stress oxidizes lipids, DNA, proteins, and various cellular macromolecules, and produces ROS which eventually results in oxidative damage [45]. Non-enzymatic antioxidants, for instance, proteins, betalains, carotenoids, and polyphenolic compounds in plants, are proficient in ROS detoxification [46]. So, the consumption of salt-induced plants can act as a reserve of powerful antioxidants in human health promotion. Consumption of these compounds provides outstanding compensations to our daily diet as a result of efficient quenching of ROS and defense against several ailments, for instance, cardiovascular ailments, cataracts, cancer, atherosclerosis, emphysema, retinopathy, arthritis, and neuron-destroying ailments [7].

Although leafy vegetables are sensitive to many stresses, amaranths are leafy vegetables broadly adjusted to diverse stresses, for instance, sodium chloride [47,48] and water deficits [49,50], and has various utilities. Various aspects, for instance, biological, physiological, ecological, biochemical, environmental, and evolutionary processes, rapidly augment the quality and contents of natural antioxidants under salinity stress [51]. Inadequate information exists on the effects of salt stress on pigments, minerals, and polyphenolic profiles in leafy vegetables. The literature has shown a salt-induced decrease in chlorophylls and increases in AsA, Folin–Ciocalteu reducing capacity, beta-carotene (BC), total flavonoids (TF), and ARC in *Cichorium spinosum* [52]. Various salt concentrations boosted the carotenoids in the sprouts of *Fagopyrum esculentum* compared to a control [53]. ARC, AsA, Folin–Ciocalteu reducing capacity, BC, and TF of purslane were ameliorated under salt stress [54]. Folin–Ciocalteu reducing capacity and ARC were increased under salt stress in barley [55]. Sodium chloride’s impact on nutrients, AsA, Folin–Ciocalteu reducing capacity, BC, TF, pigments, polyphenolic profiles, and ARC in Lalshak was studied for the first time. We hypothesized that, similar to the above-mentioned crops, nutrients, phytochemicals, pigments, polyphenolic profiles, and ARC could be increased under various salinity stresses due to abiotic stress tolerance. We screened high-ARC and high-yielding genotypes (accession LS6) based on our preceding studies [56,57,58,59,60,61]. Based on the above hypothesis, the current study aimed to estimate the influence of salt stress in a selected Lalshak genotype on nutrients, AsA, Folin–Ciocalteu reducing capacity, BC, TF, pigments, polyphenolic profiles, and ARC.

## 2. Materials and Methods

### 2.1. The Experimental Site, Plant Materials, and Experimental Conditions

Earlier, we evaluated 120 genotypes [53,54,55,56,57,58] from our departmental collection. From these studies, we selected a high-yield, high-ARC genotype (accession LS6). The seeds were sown in plastic pots at Bangabandhu Sheikh Mujibur Rahman Agricultural University (AEZ-28, 24°23′ north latitude, 90°08′ east longitude, 8.4 m.s.l.) following a randomized complete block design (RCBD) in four replicates. Recommended fertilizer doses were used. Four salt treatments, namely 100, 50, 25, and 0 mM (control), were used in the study. For proper establishment and vigorous growth of seedlings, pots were regularly irrigated through fresh water for 10 d. On the 11th day, we imposed salt treatments and continued up to 30 d (edible stage). Pots were irrigated with 100, 50, and 25 mM saline water (NaCl) and freshwater once a day. At 30 d, Lalshak leaves were harvested.

### 2.2. Chemicals

Solvent: MeOH and acetone. Reagents: NaOH, AsA, HClO_4_, cesium chloride, HNO_3_, Trolox, H_2_SO_4_, dithiothreitol (DTT), Sb, 2, 2-dipyridyl, standard flavonoid compounds, HPLC-grade acetonitrile and acetic acid, rutin, gallic acid, DPPH, sodium carbonate, Folin-Ciocalteu reagent, ABTS^+^, potassium acetate, aluminum chloride hexahydrate, and potassium persulfate. All solvents and reagents were bought from Merck (Darmstadt, Germany) and Kanto Chemical Co., Inc. (Tokyo, Japan).

### 2.3. Determination of Mineral Composition

We dried the leaves in an oven for 24 h at 70 °C. We digested ground leaves with HNO_3_ and HClO_4_ to determine the mineral elements [62,63]. We digested a 0.5 g leaf sample with 400 mL HNO_3_ (65%), 40 mL HClO_4_ (70%), and 10 mL H_2_SO_4_ (96%). We read the absorbance at 589 (Na), 213.9 (Zn), 258.056 (S), 285.2 (Mg), 248.3 (Fe), 76 6.5 (K), 422.7 (Ca), 279.5 (Mn), 880 (P), 324.8 (Cu), 313.3 (Mo), and 430 (B) nm wavelengths using a Hitachi atomic absorption spectrophotometer (Tokyo, Japan). We expressed macro- and microelements in mg g^−1^ and µg g^−1^ DW.

### 2.4. Determination of Chlorophylls and Carotenoids

The Lalshak leaves were extracted in 80% acetone to estimate total chlorophyll, chlorophyll *b*, carotenoids, and chlorophyll *a* [64,65,66,67]. A spectrophotometer (Hitachi, U-1800, Tokyo, Japan) was used to read the absorbance at 663 nm for chlorophyll *a*, 646 nm for chlorophyll *b*, and 470 nm for carotenoids. Data were expressed as µg chlorophyll per g fresh weight (FW) and mg carotenoids per 100 g FW.

### 2.5. Betacyanins and Betaxanthins Content Measurement

The Lalshak leaves were extracted in 80% methyl alcohol containing 50 mM ascorbate to measure betacyanins and betaxanthins [68,69,70]. A spectrophotometer (Hitachi, U-1800, Tokyo, Japan) was used to measure the absorbance at 540 nm for betacyanins and 475 nm for betaxanthins. The results were expressed as the nanogram betanin equivalent per gram FW for betacyanins and nanogram indicaxanthin equivalent per gram FW for betaxanthins.

### 2.6. Estimation of BC

Fresh leaves (500 mg) were thoroughly mixed with 80% acetone (10 mL) using a mortar and pestle. We determined BC by centrifuging the mixture for 3–4 min at 10,000× *g* [71,72,73,74]. We separated the filtrate in a flask and maintained the final volume of 20 mL. We measured the absorbance using a spectrophotometer (Tokyo, Japan) at 480 and 510 nm. Finally, we calculated BC as mg 100 g^−1^ FW.

### 2.7. Estimation of AsA

AsA and DHA were determined from fresh leaves. DHA was reduced to AsA by pre-incubating the sample using dithiothreitol (DTT). Fe^3+^ was converted to Fe^2+^ with the reduction of AsA. Fe^2+^ complexes were formed by reacting Fe^2+^ and 2, 2-dipyridyl [75,76]. We took the optical density of the complexes using a Hitachi spectrophotometer (Tokyo, Japan) at 525 nm. Finally, we calculated AsA as mg 100 g^−1^ FW.

### 2.8. Sample Extraction and Determination of Folin–Ciocalteu Reducing Capacity, TF, and ARC

To avoid direct sunshine, we dried leaves in a shady place. We extracted both the ground dried and fresh leaves (30 d) separately with a mortar and pestle. Folin–Ciocalteu reducing capacity was estimated from fresh leaves, whereas ARC and TF contents were estimated from dried leaves. Exactly 0.25 g samples were combined with 10 mL MeOH (90%) in a tightly capped bottle. We placed the mixture in a shaker (Tokyo, Japan) at 60 °C for 1 h. For Folin–Ciocalteu reducing capacity, ARC, and TF estimation, we stored the final filtrate until use. Folin–Ciocalteu reducing capacity and TF were estimated by the Folin-Ciocalteu reagent and AlCl_3_ colorimetric methods, respectively [77]. We used a spectrophotometer (Hitachi, Tokyo, Japan) to read the absorbance at 415 and 760 nm. Folin–Ciocalteu reducing capacity and TF were measured using gallic acid and rutin standard curves as gallic acid and rutin equivalent μg GAE g^−1^ of FW and μg RE g^−1^ DW. The ARC was estimated by radical degradation by DPPH and ABTS^+^ assay [78,79]. We measured the inhibition % of ABTS^+^ and DPPH equivalent to the control using the equation:ARC (%) = (Ab − AS/Ab) × 100
where Ab represents the blank sample absorbance (10 µL and 150 μL MeOH for ARC (ABTS) and DPPH, respectively, as a substitute of leaf extract), and AS is the absorbance of the sample. Finally, we calculated ARC as μg Trolox equivalent g^−1^ DW.

### 2.9. Sample Extraction and Determination of Polyphenolic Profiles by HPLC

Exactly 1 g of fresh leaves was extracted in 10 mL MeOH (80%) comprising ascorbate (1%). We homogenized the mixture thoroughly and placed it in a 50 mL test tube (tightly capped). Then, we placed the test tube in a shaker (Scientific Industries Inc., New York, NY, USA) at 400 rpm for 15 h. The mixture was filtered using a 0.45 µm filter (Springfield, MA, USA) and centrifuged at 10,000× *g* for 15 min. We estimated polyphenolic compounds from the filtrate. We repeated all extractions 3 times. For the HPLC determination of polyphenolic compounds, we followed the method of Sarker and Oba [50]. We equipped Shimadzu HPLC equipment (Kyoto, Japan) with a degasser, detector, and binary pump. A CTO-10 AC (STR ODS-II, 150 × 4.6 mm, 5 µm; Shinwa Chemical Industries, Ltd., Kyoto, Japan) column was used for the separation of polyphenolic compounds. We pumped Solvent A (acetic acid 6% (*v*/*v*) in water) and Solvent B (acetonitrile) at 1 mL/min for 70 min. We used a gradient program to run the HPLC system with 0–15%, 15–30%, 30–50%, and 50–100% acetonitrile for 45, 15, 5, and 5 min. We maintained the column temperature of 35 °C with an injection volume of 10 µL. We set a Shimadzu SPD-10Avp UV–vis detector at 280, 360, and 370 nm to continuously monitor polyphenolic compounds. We identified the compound by comparing the retention time and UV–vis spectra with their respective standards. Finally, we calculated polyphenolic compounds as µg g^−1^ FW.

### 2.10. Quantification of Polyphenolic Compounds

We quantified each polyphenolic compound using the corresponding standards of calibration curves. We prepared stock solutions (100 mg/mL) by dissolving 9 polyphenolic compounds with 80%MeOH. We quantified polyphenolic compounds using standard curves (10, 20, 40, 60, 80, and 100 μg/mL) with external standards. Co-chromatography of samples’ retention times spiked with commercially available standards. We identified and matched the polyphenolic compounds utilizing UV spectral characteristics.

### 2.11. Statistical Analysis

To obtain a replication mean, we averaged each treatment from all the sample data of a trait [80,81,82,83]. We biometrically and statistically analyzed the mean data of various traits [84,85,86,87]. Statistix 8 software was used to analyze the data to obtain an analysis of variance (ANOVA) [88,89,90]. Duncan’s multiple range test (DMRT) at a 1% level of probability was used to compare the means. The results were reported as the mean ± SD of four separate replicates.

## 3. Results and Discussion

### 3.1. Influence of Sodium Chloride Stress on Color Parameters and Pigments

Figure 1 and Figure 2 represent the color parameters and pigments under different sodium chloride stresses. Preference, choice making, and acceptability of the product mostly depend on leaf color, which contributes significantly to the choice of consumers. It is a key indicator for evaluating the ARC of leafy vegetables [91]. The LS6 accession had high positive a* and b* values, indicating the presence of abundant red and yellow color pigments (betaxanthins, carotenoids, betacyanins, anthocyanins, and betalains). Our obtained results corroborated the results of Colonna et al. [91]. Betacyanins, chroma, L*, carotenoids, betaxanthins, betalains, a*, and b* values were progressively augmented in the order: Control < 25 < 50 < 100 mM salt stress. In contrast, total chlorophyll, chlorophyll *b*, and chlorophyll *a* content were drastically reduced in the order: Control > 25 > 50 > 100 mM salt stress. Carotenoids, L*, chroma, b*, betacyanins, betaxanthins, betalains, and a* were augmented by 14%, 0%, 1%, 1%, 2%, 4%, 1%, 4%; 28%, 32%, 3%, 7%, 7%, 6%, 5%, 3%, 11%; and 59%, 6%, 14%, 16%, 10% 10%, 8%, 22% under 25; 50; and 100 mM salt concentrations, respectively. In contrast, chlorophyll *b*, chlorophyll *a,* and total chlorophyll content were reduced by 2%, 6%, 4%; 9%, 9%, and 9%; and 17%, 19%, 18%, respectively, compared to control conditions (Figure 3). Petropoulos et al. [52] reported that the chlorophylls of *Cichorium spinosum* were drastically reduced with an increment in sodium chloride stress.

Sodium chloride stress affected plant growth and development through decreasing stomatal conductivity, which restricts CO_2_ influx to leaves and causes osmotic stress in plants, reduction in water potential, and unfavorable CO_2_/O_2_ ratios in chloroplasts, reducing photosynthesis. Lim et al. [53] reported that different salt concentrations augmented carotenoid content. They observed the highest increment (two-fold) in carotenoids under 50 and 100 mM salt concentrations in comparison to the control conditions. Alam et al. [54] observed both stimulation and reduction in carotenoid content in salt-stressed purslane. To regulate plant development under sodium chloride stress, the plant accelerates the mevalonic acid pathway for the biogenesis of abscisic acid from carotenoids. Thus, sodium chloride stress enhances the synthesis of carotenoids to accelerate the mevalonic acid pathway [47]. The decline in pigment for photosynthesis under salt stress is also linked with the oxidation of chlorophyll pigment through free radicals, interference of salt ions with pigment–protein complexes [92], chloroplast disruption, and increased activity of chlorophyllase enzymes responsible for the degradation of chlorophylls [93]. The presence of betalain pigments (betaxanthin and betacyanin) may act as an antioxidant and absorb radiation significantly to protect against excessive harmful light in the chloroplasts. These findings were corroborative to the findings of Jain et al. [94]. In *Disphyma australe*, they reported that salt-induced plants with increased betalains exhibited more tolerant physiology through the production of less H_2_O_2_, faster recoveries of PSII, and increased rates of assimilation of CO_2_, and photochemical quenching, photochemical quantum yields, and water-use efficiencies. Moreover, betalains (betacyanins and betaxanthins) protect the chloroplasts from salinity stress by scavenging reactive oxygen species in thylakoids [95] and through faster recoveries of PSII, photochemical quenching, and photochemical quantum yields [94].

### 3.2. Sodium Chloride Impact on Minerals (Macroelements and Microelements)

Macroelements and microelements in Lalshak are presented in Figure 4 and Figure 5. The studied Lalshak demonstrated copious macroelements and microelements, which corroborated with the results of Shukla et al. [96], who reported very high levels of minerals in open-field-grown *A. tricolor*. Lalshak has greater iron and zinc compared to the leaves of cassava [97] and beach peas [98]. The previous study showed copious amounts of Mn, Fe, Cu, and Zn in different *A.* spp. [99]. They demonstrated greater levels of copper and iron in different *A.* spp., which were superior to kale, and Zn levels of different *A.* spp. were also superior to spinach, kale, and black nightshade. At 100 mM salt concentration, the maximum calcium, magnesium, sulfur, iron, manganese, copper, zinc, sodium, molybdenum, and boron contents were noted, while at control conditions, the minimum calcium, magnesium, manganese, zinc, sodium, and boron contents were displayed. Similarly, under control and 25 mM salt stress conditions, the minimum sulfur, iron, copper, and molybdenum contents were detected. Calcium, magnesium, manganese, zinc, sodium, and boron contents were gradually increased in the order: Control < 25 < 50 < 100 mM salt concentrations. Inversely, potassium and phosphorus contents extremely declined in the order: Control > 25 > 50 > 100 mM salt concentrations.

In 25, 50, and 100 mM salt concentrations, calcium, magnesium, sulfur, iron, manganese, copper, zinc, sodium, molybdenum, and boron contents were increased by 11%, 10%, −7.5%, −1%, 13%, 3%, 10%, 3%, 5%, and 4%; 24%, 25%, 16%, 6%, 29%, 36%, 46% 55%, 66%, and 20%; and 30%, 44%, 30%, 40%, 57%, 64%, 69%, 94%, 18%, and 45%, respectively, compared to control conditions (Figure 6). In 25, 50, and 100 mM salt concentrations, potassium and phosphorus contents were reduced to 9%, 20%, and 30%, and 1%, 19%, and 30%, respectively, compared to control conditions (Figure 6).

Petropoulos et al. [52] detected a progressive increment in minerals, which corroborated the current findings. Petropoulos et al. [52] reported a sharp rise in calcium, iron, magnesium, manganese, sodium, and zinc content and a reduction in potassium content in *C. spinosum*. They reported that salt treatment and fertilizer application could be the cause for the enhancement of sodium content and recommended that, to cope with the adverse effects of salinity, the species accumulate sodium. The iron content of Lalshak was statistically similar to the value of control and 25 mM salt concentration conditions, while iron content was gradually increased under 50 and 100 mM salt concentration conditions by 12% and 62%, respectively. At 25 mM salt concentration, the minimum sulfur content was attained, which fluctuated noticeably from the control conditions. The sulfur content was progressively increased at 50 and 100 mM salt concentrations by 20% and 51%, respectively (Figure 6). Menezes et al. [100] and Odjegba and Chukwunwike [101] reported a similar increase in Na^+^ and a reduction in K^+^ content at different salt concentrations in *A. cruentus* and *A. hybridus*, respectively. Koksal et al. [102] showed that Ca^2+^ and Mg^2+^ increased in the roots and shoots as the salinity stress increased, while the K^+^ concentration decreased in marigolds, which corroborated the present findings. Non-specific ion uptake in salt-induced cells raises the concentration of Na^+^ ions. In salt-tolerant plants, two main mechanisms, namely salt exclusion and sequestration, are identified to maintain cytosolic Na^+^ levels appropriately [103]. In many plant species, the main physiological mechanism of salt tolerance is the uptake of selective K^+^ against Na^+^ [104].

### 3.3. Influence of Sodium Chloride on Phytochemicals

The Folin–Ciocalteu reducing capacity, BC, AsA, TF, and ARC varied noticeably at different sodium chloride concentrations (Figure 7).

Sodium chloride concentrations gradually increased Folin–Ciocalteu reducing capacity, BC, AsA, TF, and ARC in the following order: Control < 25 < 50 < 100 mM salt concentrations. BC, AsA, Folin–Ciocalteu reducing capacity, TF, and ARC (DPPH and ABTS^+^) under 25, 50, and 100 mM salt concentration conditions were progressively increased by 14%, 6%, 10%, 6%, 6%, and 5%; 37, 21%, 23%, 22%, 20%, and 19%; and 52%, 55%, 57%, 41%, 38%, and 40%, respectively, compared to control conditions (Figure 8).

BC, AsA, Folin–Ciocalteu reducing capacity, TF, and ARC (DPPH and ABTS^+^) of Lalshak were maximized at 100 mM salt stress. On the contrary, minimum BC, AsA, Folin–Ciocalteu reducing capacity, TF, and ARC (DPPH and ABTS^+^) values were noted under control conditions. Petropoulos et al. [52] described the salinity-mediated increases in AsA, Folin–Ciocalteu reducing capacity, BC, TF, and ARC in *Cichorium spinosum*. Different concentrations of sodium chloride boosted the carotenoid content in buckwheat sprouts compared to a control [53]. Alam et al. [54] reported salt-induced enhancement of TF and ARC in purslane. Ahmed et al. [55] demonstrated a salinity-mediated rise in Folin–Ciocalteu reducing capacity and ARC in barley. Salinity stress creates excessive ROS which eventually causes oxidative stress in plants. To cope with oxidative damage, plants accumulate several secondary metabolites and non-enzymatic antioxidant compounds, such as BC, AsA, polyphenols, flavonoids, and antioxidant enzymes. BC plays a main protective role in photosynthetic tissue by protecting it from oxidative damage, preventing the generation of singlet oxygen and direct scavenging of triplet chlorophyll [105]. Non-enzymatic antioxidants, such as ascorbic acid, phenolics, and flavonoids, play an important role in reducing oxidative stress and cellular ROS homeostasis regulation in plants [106].

### 3.4. Response of Sodium Chloride Stress on Polyphenolic Compounds

The HPLC-identified polyphenolic profile values of Lalshak (accession LS6) under four salt stresses were collated with polyphenolic compounds using the respective peaks of the compounds (Table 1). Figure 9 designates the identified polyphenolic profiles of the Lalshak genotype under four salt stresses. Nine polyphenolic profiles including six flavonols, namely quercetin, rutin, iso-quercetin, hyperoside, kaempferol, and myricetin; one flavanol (catechin); one flavone (apigenin); and one flavanone (naringenin) were identified in adequate quantities in Lalshak leaves. We identified six polyphenolic compounds (iso-quercetin, kaempferol, myricetin, catechin, apigenin, and naringenin) for the first time in this genotype. Across polyphenolic profiles, rutin is the most preponderant flavonoid compound in Lalshak followed by quercetin, naringenin, and myricetin (Figure 9). Khanam et al. [107] and Khanam and Oba [108] reported three flavonoids (quercetin, rutin, and hyperoside) in amaranths.

Abiotic stresses such as salinity generate various ROS, such as H_2_O_2_, superoxide, hydroxyl radical, singlet oxygen, etc., and cause oxidative damage in plants which, finally, can oxidize lipids, DNA, proteins, and various cellular macromolecules. To cope with oxidative damage, plants accumulate non-enzymatic antioxidant compounds, such as polyphenols, flavonoids, and antioxidant enzymes. Generally, the accumulation of polyphenols that possess antioxidant properties is stimulated in response to ROS increases under biotic and abiotic stresses. They are plentiful and present in plant tissues [109]. Polyphenols can chelate transition-metal ions, can directly scavenge molecular species of active oxygen, and may quench lipid peroxidation by trapping the lipid alkoxyl radical. Furthermore, flavonoids and phenylpropanoids are oxidized by peroxidase and act in the H_2_O_2_-scavenging, phenolic/AsA/POD system. Antioxidant activity is the combined results of all enzymatic and non-enzymatic antioxidant activity in natural and/or biotic/abiotic stress. Tolerant plant genotypes usually have a better antioxidant content to protect them from oxidative stress by maintaining high antioxidant enzyme and antioxidant molecule activities under stress conditions. Antioxidants protect the cells from free radicals and, therefore, have been considered as a method to improve plant defense responses [110]. Antioxidant activity has a crucial role in maintaining the balance between the production and scavenging of free radicals [111].

Salt stress progressively augmented all flavonoid compositions. At 100 mM salt stress, all flavonoid compounds showed maximum contents, while the lowest flavonoid contents were recorded from the control treatment. Quercetin, rutin, hyperoside, myricetin, and naringenin were progressively augmented in the following order: Control < 25 < 50 < 100 mM salt stress. From control to 100 mM salt stress conditions, quercetin, rutin, hyperoside, myricetin, and naringenin ranged from 7.35 to 18.63, 14.62 to 32.47, 3.35 to 7.36, 7.48 to 15.48, and 9.14 to 16.58 µg g^−1^ FW, respectively (Figure 9). From control to 100 mM salt concentration conditions, quercetin, rutin, hyperoside, myricetin, and naringenin were sharply and remarkably augmented by 16%, 110%, and 153%; 21%, 56%, and 112%; 19%, 95%, and 120%; 9%, 57%, and 107%; and 14%, 36%, and 81% (Figure 10).

Iso-quercetin did not augment between control and 25 mM salt stress conditions; however, when increasing salt concentration from 25 to 100 mM, this compound sharply increased with an increase in salt concentration with a range from 6.01 to 8.96 µg g^−1^ FW. Apigenin sharply increased from control to 50 mM salt stress conditions with a range from 6.37 to 7.97 µg g^−1^ FW. However, when increasing salt concentration from 50 to 100 mM, the apigenin concentration statistically remained constant. Kaempferol and catechin ranged from 7.88 to 10.86 and 2.88 to 5.36 µg g^−1^ FW. These two compounds had statistical similarity between the control and 25 mM salt stress conditions and between 50 and 100 mM salt stress conditions; however, these two compounds were remarkably augmented from the control condition or 25 to 50 or 100 mM salt stress conditions (82%) (Figure 9 and Figure 10).

Among the four groups of polyphenolic profiles, the flavonols group is the most plentiful in Lalshak compared to other groups, followed by flavanones. Polyphenolic groups in Lalshak were in the order: flavonols > flavanones > flavones > flavanols (Figure 11). All polyphenolic portions were abruptly increased under salt stress. All polyphenolic portions displayed maximum concentrations under 100 mM salt concentrations, although the control had minimum polyphenolic portions. From control to 100 mM salt concentration, flavonols, flavones, flavanols, flavanones, and total polyphenols ranged from 46.66 to 93.76, 6.37 to 8.06, 2.88 to 5.36, 9.14 to 16.58, and 65.05 to 123.76 µg g^−1^ FW, respectively (Figure 11).

Flavonols, flavanones, and total polyphenols were progressively augmented in the order: Control < 25 < 50 < 100 mM salt concentrations, while flavones and flavanols were progressively augmented in the order: Control = 25 < 50 = 100 mM salt concentrations (Figure 11). In 25, 50, and 100 mM salt concentration conditions, flavonols, flavones, flavanols, flavanones, and total polyphenols were predominately augmented by 12%, 60%, and 101%; 14%, 25%, and 27%; 5%, 82%, and 86%; 14%, 36%, and 81%; and 12%, 5%, and 90%, respectively, compared to control conditions (Figure 12).

### 3.5. The Coefficient of Correlation Study

The coefficient of correlation among BC, AsA, TF, Folin–Ciocalteu reducing capacity, ARC (DPPH), and ARC (ABTS^+^) are shown in Table 2. BC showed significant associations with AsA, TF, Folin–Ciocalteu reducing capacity, ARC (DPPH), and ARC (ABTS^+^). This indicated that the augmentation of BC is predominately related to the enhancement of AsA, TF, Folin–Ciocalteu reducing capacity, ARC (DPPH), and ARC (ABTS^+^). Similarly, AsA exhibited a significant inter-relationship with TF, Folin–Ciocalteu reducing capacity, ARC (DPPH), and ARC (ABTS^+^). Both BC and AsA had significant and strong contributions to the ARC of the genotype.

TF, Folin–Ciocalteu reducing capacity, ARC (DPPH), and ARC (ABTS^+^) were significantly correlated with each other. Gharibi et al. [112] observed a positive association among total Folin–Ciocalteu reducing capacity, TF, and ARC in *Achillea* species. Alam et al. [54] also reported a significant correlation among carotenoids, Folin–Ciocalteu reducing capacity, AsA, BC, and TF with ARC (FRAP) in salt-stressed purslane. Significant positive associations of AsA, Folin–Ciocalteu reducing capacity, BC, TF, ARC (DPPH), and ARC (ABTS^+^) signifies the strong antioxidant potential of TF and Folin–Ciocalteu reducing capacity of the genotype. Likewise, significant positive correlations between ARC (DPPH) and ARC (ABTS^+^) confirmed the validation of the antioxidant potential of the genotype by estimation of ARC using two different methods.

## 4. Conclusions

Sodium chloride stress remarkably augmented a*, calcium, L*, AsA, magnesium, b*, ARC (DPPH), sulfur, TF, iron, BC, manganese, ARC (ABTS^+^) copper, zinc, sodium, Folin–Ciocalteu reducing capacity, molybdenum, boron, chroma, polyphenolic profiles, and pigments such as betacyanins, betaxanthins, betalains, and carotenoids of Lalshak leaves. All mineral contents, AsA, Folin–Ciocalteu reducing capacity, BC, TF, pigments, polyphenolic profiles, and ARC of Lalshak leaves under 50 and 100 mM salt concentrations were much higher in comparison to the control conditions. It could be used as a valuable food for human diets with health benefits. Salt-treated Lalshak leaves had abundant minerals, AsA, Folin–Ciocalteu reducing capacity, BC, TF, pigments, polyphenolic profiles, and ARC. Pigments, AsA, Folin–Ciocalteu reducing capacity, BC, TF, polyphenolic compounds, and ARC quench ROS; thus, Lalshak could be beneficial for human health via its potent antioxidant activities. Moreover, sodium chloride-enriched Lalshak provided outstanding quality in the final product in terms of nutrients, pigments, polyphenolic profiles, and ARC. We can cultivate it as an encouraging alternative vegetable in salt-prone zones of the world.

## Figures and Tables

**Figure 1 antioxidants-12-00173-f001:**
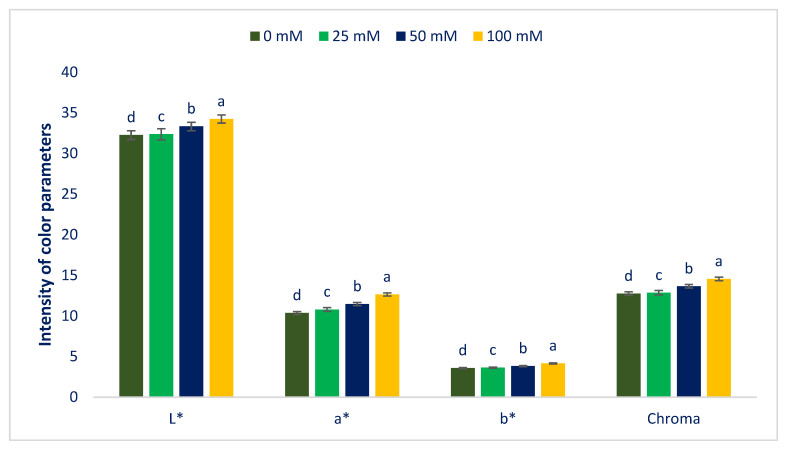
Color attributes of Lalshak as influenced by salt stress (*n* = 6). L* = Lightness; a* = Redness/greenness; b* = Yellowness/blueness. Dissimilar letters in the bars varied significantly by Duncan’s Multiple Range Test (DMRT) (*p* < 0.01).

**Figure 2 antioxidants-12-00173-f002:**
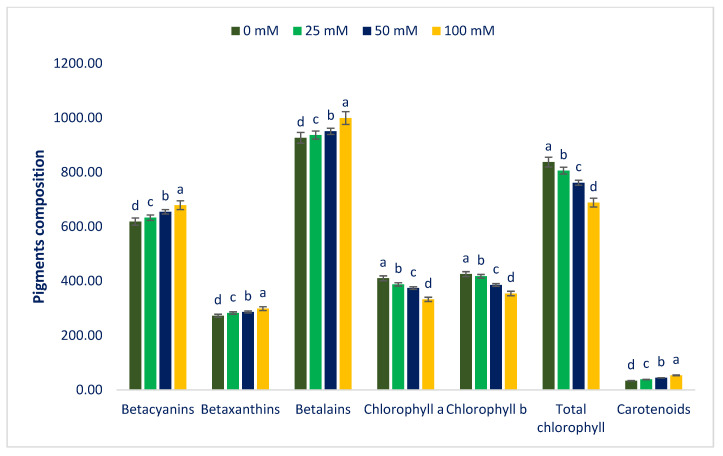
Pigments of Lalshak as influenced by salt stress (*n* = 6). Dissimilar letters in the bars varied significantly by DMRT (*p* < 0.01).

**Figure 3 antioxidants-12-00173-f003:**
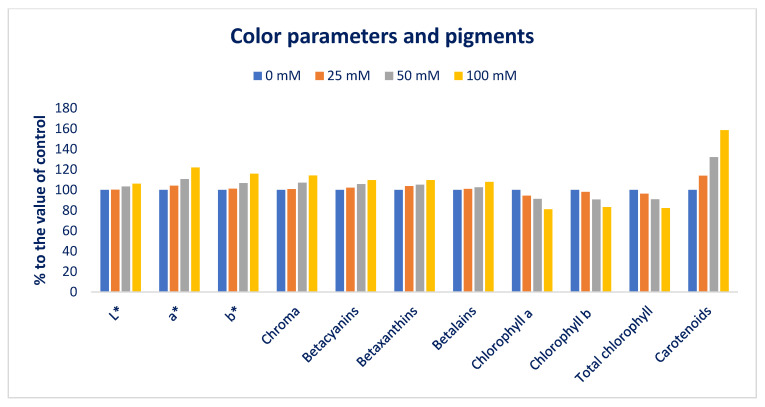
The salt-induced increment in color attributes and pigments (%) over control in Lalshak.

**Figure 4 antioxidants-12-00173-f004:**
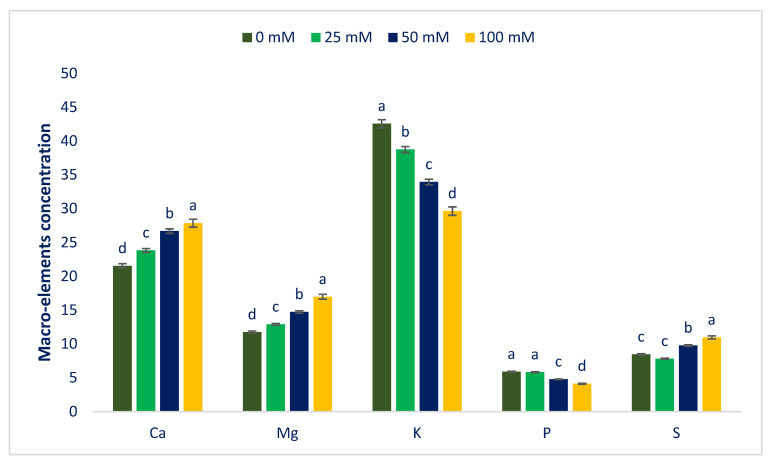
Macroelements (mg g^−1^ DW) of Lalshak as influenced by salt stress (*n* = 6). Dissimilar letters in the bars varied significantly by DMRT (*p* < 0.01).

**Figure 5 antioxidants-12-00173-f005:**
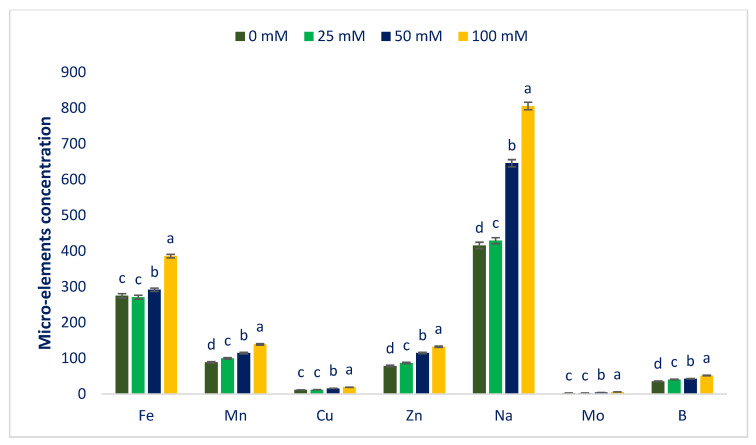
Microelements (µg g^−1^ DW) of Lalshak as influenced by salt stress (*n* = 6). Dissimilar letters in the bars varied significantly by DMRT (*p* < 0.01).

**Figure 6 antioxidants-12-00173-f006:**
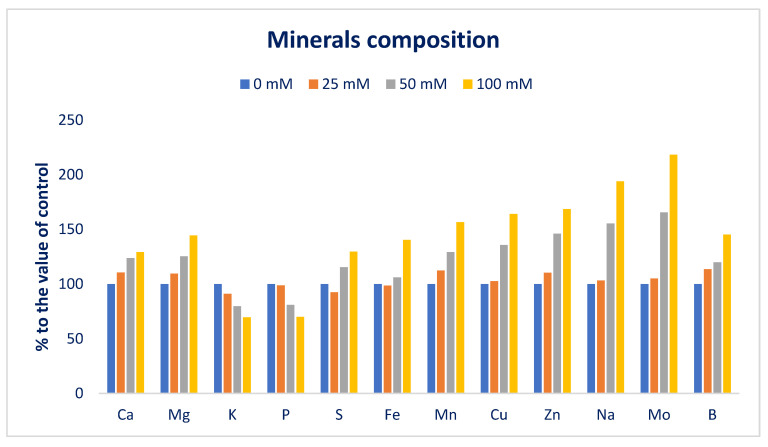
Comparison of the salt-induced increment in mineral composition (%) over control in Lalshak.

**Figure 7 antioxidants-12-00173-f007:**
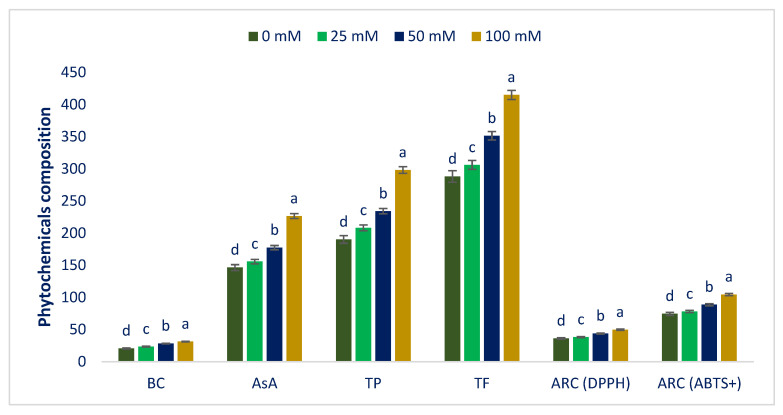
Response of phytochemical composition of Lalshak under salt stress (*n* = 6). Dissimilar letters in the bars varied significantly by DMRT (*p* < 0.01).

**Figure 8 antioxidants-12-00173-f008:**
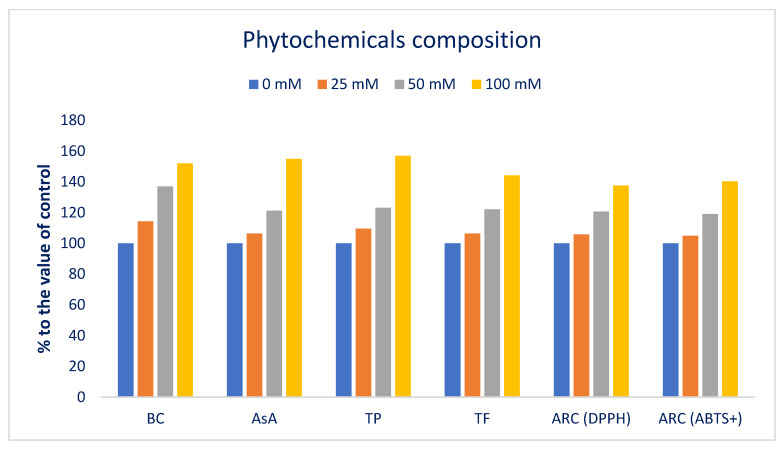
Comparison of the salt-induced increment in phytochemical composition (%) over control in Lalshak.

**Figure 9 antioxidants-12-00173-f009:**
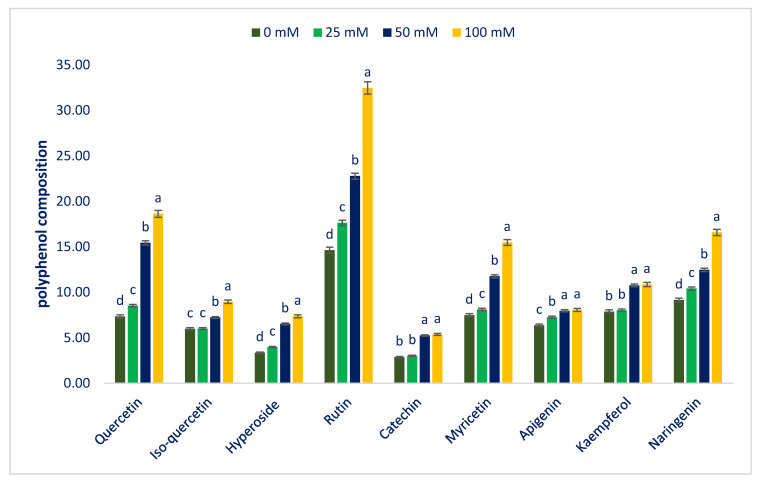
Impact on polyphenolic compounds of Lalshak under salt stress (*n* = 6). Dissimilar letters in the bars varied significantly by DMRT (*p* < 0.01).

**Figure 10 antioxidants-12-00173-f010:**
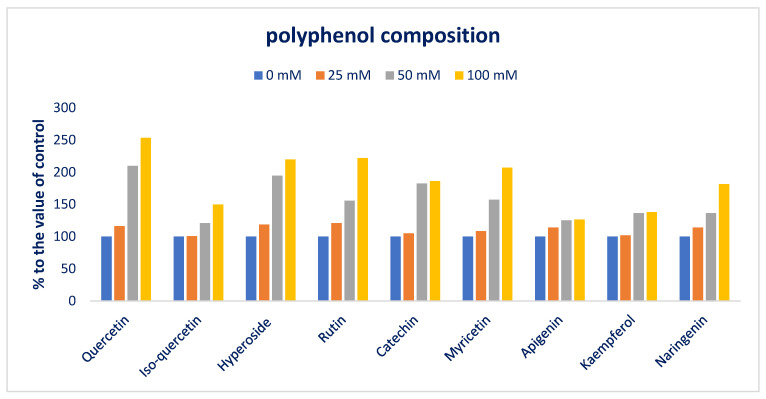
Comparison of the salt-induced increment in polyphenolic compounds (%) over control in Lalshak.

**Figure 11 antioxidants-12-00173-f011:**
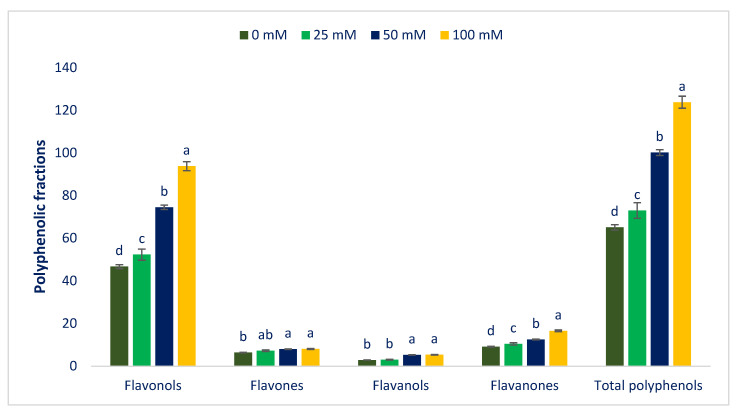
Impact on polyphenolic fractions of Lalshak under salt stress (*n* = 6). Dissimilar letters in the bars varied significantly by DMRT (*p* < 0.01).

**Figure 12 antioxidants-12-00173-f012:**
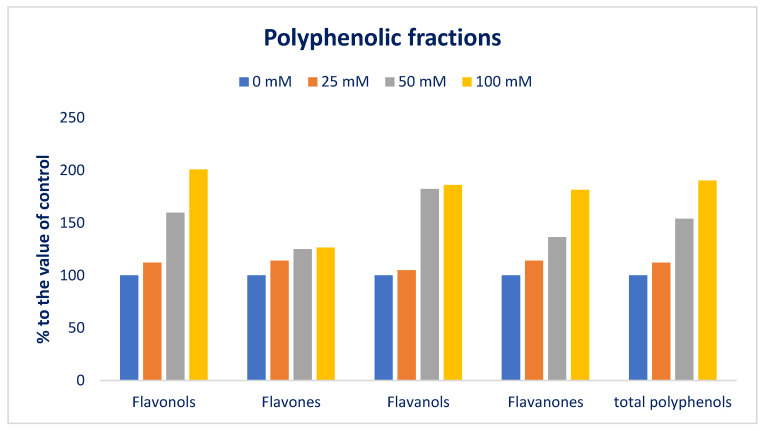
Comparison of the salt-induced increment polyphenolic fractions (%) over control in Lalshak.

**Table 1 antioxidants-12-00173-t001:** Retention time (Rt), wavelengths of maximum absorption in the visible region (λ_max_), mass spectral data, and tentative identification of polyphenolic compounds in Lalshak.

Rt (min)	λ_max_ (nm)	Molecular Ion [M − H]^−^ (*m*/*z*)	MS^2^ (*m*/*z*)	Identity of Tentative Compounds
4.6	370	626.2468	626.3216	Myricetin-3-O-rutinoside
7.5	370	301.0348	301.2267	2-(3,4-dihydroxy phenyl)-3,5,7-trihydroxychromene-4-one
15.4	370	270.2344	270.3221	4′,5,7-Trihydroxyflavone, 5,7-Dihydroxy-2-(4-hydroxyphenyl)-4-benzopyrone
17.8	370	593.4253	593.3687	kaempferol-3-O-rutinoside
23.9	280	290.2463	290.1238	(2R-3S)-2-(3,4-dihydroxyphenyl)-3,4-dihydro-2-chromene-3,5,7-triol
26.7	280	271.0622	271.2448	Naringenin
53.0	360	609.3874	609.4265	Quercetin-3-O-rutinoside
53.3	360	463.4358	463.5125	Quercetin-3-O-galactoside
54.3	360	463.2875	463.3124	Quercetin-3-O-glucoside

**Table 2 antioxidants-12-00173-t002:** The correlation coefficient for Folin–Ciocalteu reducing capacity (FCRC), AsA, BC, TF, and ARC (DPPH and ABTS^+^) in Lalshak.

	AsA	FCRC	TF	ARC (DPPH)	ARC (ABTS^+^)
BC	0.82 **	0.81 **	0.96 **	0.79 *	0.84 *
AsA		0.88 **	0.91 **	0.92 **	0.96 **
FCRC			0.81 *	0.93 **	0.94 **
TF				0.82 *	0.78 *
ARC (DPPH)					0.97 **

* and **: significant at 5% and 1% level (*n* = 6).

## Data Availability

The data that are recorded in the current study are available in all of the tables and figures of the manuscript.

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
