# Peer review of "Salinity Stress Ameliorates Pigments, Minerals, Polyphenolic Profiles, and Antiradical Capacity in Lalshak"

_antioxidants, 2023, doi:10.3390/antiox12010173_

Round 1

Reviewer 1 Report (New Reviewer)

Overall the manuscript is well written and it can be accepted for publcation after these minor corrections

In the introduction, you should state your hypothesis explicitly and then set the stage for the following paragraph.

A careless approach was taken in the writing of the figure ligands.

Rearrange the information, and give the concluding section a thorough reading through while also editing it.

Author Response

Reviewer 1 (Round 1)

Comments: Overall the manuscript is well written and it can be accepted for publcation after these minor corrections

Author response: Thank you for allowing us the opportunity to submit our revised manuscript for publication in the Antioxidants Journal of MDPI. We appreciate the time and effort you have taken to improve our manuscript. We are also thankful to the honorable reviewer for the positive decision to publish in Antioxidants. We revised our manuscript following your point-by-point comments and suggestions for substantial improvement. We hope that this revised version satisfies you to take the final decision.

Comments: In the introduction, you should state your hypothesis explicitly and then set the stage for the following paragraph.

Author response: Thank you for your comments. We have revised it following your suggestion.

Comments: A careless approach was taken in the writing of the figure ligands.

Author response: Thank you for your comments. We have revised the figure legends.

Comments: Rearrange the information, and give the concluding section a thorough reading through while also editing it.

Author response: Thank you for your comments. We have included a few concluding sentences at the end of the conclusion.  

Reviewer 2 Report (New Reviewer)

This article entitled “Salinity stress Ameliorates Pigments, Minerals, Polyphenolic Profiles, and Antiradical Capacity in Lalshak”. In the present study, the authors evaluated the response of sodium chloride stress in a selected Lalshak (A. gangeticus) genotype in terms of minerals, ascorbic acid (AsA), total polyphenols (TP), beta-carotene (BC), Total flavonoids (TF), pigments, polyphenolic profiles, and ARC.

The topic is interesting and informative. The manuscript needed to be revised before it could be published. Before recommending this article for publication, some shortcomings should be resolved.

The authors should improve the overall English of the manuscript.

Some formatting mistakes have been found. I suggest authors review the whole manuscript carefully and correct all the mistakes.

The scientific names of the species and the names of the genes must be italicized in the manuscript.

The authors should fully explain the abbreviations during the first mention in the abstract and introduction.

Line 24: Total or total? Be consistent, like total polyphenols (TP)

Line 98: treatments, such as

Line 233: HNO3: It should be HNO3, please make sure.

Line 337: respectively? Please make sure.

Line 541: didn’t: please check it to did not

Author Response

Reviewer 2 (Round 1)

Comments: This article entitled “Salinity stress Ameliorates Pigments, Minerals, Polyphenolic Profiles, and Antiradical Capacity in Lalshak”. In the present study, the authors evaluated the response of sodium chloride stress in a selected Lalshak (A. gangeticus) genotype in terms of minerals, ascorbic acid (AsA), total polyphenols (TP), beta-carotene (BC), Total flavonoids (TF), pigments, polyphenolic profiles, and ARC.

The topic is interesting and informative. The manuscript needed to be revised before it could be published. Before recommending this article for publication, some shortcomings should be resolved.

Author response: Thank you for allowing us the opportunity to submit our revised manuscript for publication in the Antioxidants Journal of MDPI. We appreciate the time and effort you have taken to improve our manuscript. We are also thankful to the honorable reviewer for the positive decision to publish in Antioxidants. We revised our manuscript following your point-by-point comments and suggestions for substantial improvement. We hope that this revised version satisfies you to take the final decision. We have resolved the shortcomings according to your suggestion.

Comments: The authors should improve the overall English of the manuscript.

Author response: Thank you for your comments. The language of the manuscript was thoroughly checked by our University English Expert Team.

Comments: Some formatting mistakes have been found. I suggest authors review the whole manuscript carefully and correct all the mistakes.

Author response: Thank you for your comments. We have reviewed the whole manuscript carefully and corrected all the mistakes.

Comments: The scientific names of the species and the names of the genes must be italicized in the manuscript.

Author response: Thank you for your comments. We have italicized the scientific names.

Comments: The authors should fully explain the abbreviations during the first mention in the abstract and introduction.

Author response: Thank you for your comments. We have explained the abbreviations during the first mention in the abstract and introduction.

Comments: Line 24: Total or total? Be consistent, like total polyphenols (TP)

Author response: Thank you for your comments. We have changed “Total” to “total”.

Comments: Line 98: treatments, such as

Author response: Thank you for your comments. We have added the comma.

Comments: Line 233: HNO3: It should be HNO3, please make sure.

Author response: Thank you for your comments. We have subscripted the number.

Comments: Line 337: respectively? Please make sure.

Author response: Thank you for your comments. We didn’t find it in line 337.

Comments: Line 541: didn’t: please check it to did not

Author response: Thank you for your comments. We have checked and retained it.

Reviewer 3 Report (New Reviewer)

In the manuscript entitled “Salinity stress Ameliorates Pigments, Minerals, Polyphenolic Profiles, 2 and Antiradical Capacity in Lalshak” presented by Umakanta Sarker et al., studied the effect of salt stress on the molecular composition of Lalshak leaves. Phytochemical composition and contents were determined. They also studied mineral composition. The work is quite consequent. The work would help to understand the effect of salinity on plant metabolisms and enrich the literature.

Thanks to its theme, this investigation is suitable for publication in journal of Antioxidants; however, there are some issues which must be revised and explained before publication.

Line 50: [3] that play an important role in human health: reference is needed for the last sentence!

Line 53: of cardiovascular disease, and lung and skin cancers. We need reference of clinical studies!

Line 56: the subject is Lalshak, however, introduction focus on Amaranth!? Authors should introduce their plant and previous works on Lalshak.

Line 70-74: So, the consumption of salt-induced plants…It is the hypothesis of the work, verbs should be in conditional….all of benefits (line 71-74) could be considered in clinical studies…are not the aim of the actual study.

Line 232: We dried the leaves in an oven at 70 °C temperature for 24 h. is this a normalized method?

Line 240: The leaves of Lalshak were extracted in 80% acetone to estimate. The extraction method should be succinctly developed.

Line 244: FW!?....for accurate comparison all data should expressed as DW. All data should be recalculated!!!!

Line 257: idem

Line 259: Fresh leaves were used to determine AsA and DHA. The sample was pre-incubated 259 by Dithiothreitol (DTT)….leaves were ground or not!!! Method should briefly described.

Line 263: FW. Change to DW and redo calculation

Line 275: Leaves were dried in a shady place to avoid direct sunshine. Did authors checked the DW of this method?

Line 299: the name and nature of the used column should mentioned.

Line 305: according to result section, authors used a MS detector….It is not detailed in this paragraph!!!

Line 307: FW: recalculation using DW

Line 323: when necessary, “significant” should be used in all comparison.

Figure 3, 6, 8, 10 and 12, Standard deviation should be used.

Line 368-374: discussion and explanation about the mechanism response of the plant should be given.

Line 425-432: C. spinosum. They stated that….. by 20% and 51%, respectively (Fig. 4).

Which physiological/molecular mechanisms are used by the plant?

Line 434: 3.3. Influence of sodium chloride on phytochemicals

The method used for TP determination give Folin–Ciocalteu reducing capacity of sample and not the TP content…..authors should compare their TP content and the content calculated from HPLC data (polyphenols)!...so

Folin–Ciocalteu reducing capacity should be used instead TP content in all sections

Line 493-500: paragraph discuss clinical results. It is not the subject of this study….

Discussion should focus on response mechanisms

Author Response

Reviewer 3 (Round 1)

Comments: In the manuscript entitled “Salinity stress Ameliorates Pigments, Minerals, Polyphenolic Profiles, 2 and Antiradical Capacity in Lalshak” presented by Umakanta Sarker et al., studied the effect of salt stress on the molecular composition of Lalshak leaves. Phytochemical composition and contents were determined. They also studied mineral composition. The work is quite consequent. The work would help to understand the effect of salinity on plant metabolisms and enrich the literature.

Thanks to its theme, this investigation is suitable for publication in journal of Antioxidants; however, there are some issues which must be revised and explained before publication.

Author response: Thank you for allowing us the opportunity to submit our revised manuscript for publication in the Antioxidants Journal of MDPI. We appreciate the time and effort you have taken to improve our manuscript. We are also thankful to the honorable reviewer for the positive decision to publish in Antioxidants. We revised our manuscript following your point-by-point comments and suggestions for substantial improvement. We hope that this revised version satisfies you to take the final decision. 

Comments: Line 50: [3] that play an important role in human health: reference is needed for the last sentence!

Author response: Thank you for your comments. We have cited a reference [4].

Comments: Line 53: of cardiovascular disease, and lung and skin cancers. We need reference of clinical studies!

Author response: Thank you for your comments. We have added 2 references [5,6].

Comments: Line 56: the subject is Lalshak, however, introduction focus on Amaranth!? Authors should introduce their plant and previous works on Lalshak.

Author response: Thank you for your comments. Lalshak is a variety of Amaranthus species (Amaranth).

Comments: Line 70-74: So, the consumption of salt-induced plants…It is the hypothesis of the work, verbs should be in conditional….all of benefits (line 71-74) could be considered in clinical studies…are not the aim of the actual study.

Author response: Thank you for your comments. We have added these sentences to clarify the importance of this research citing from the literature. We have stated the hypothesis in the last paragraph (lines 88-90). We have stated the aim of our study in lines 91-93.

Comments: Line 232: We dried the leaves in an oven at 70 °C temperature for 24 h. is this a normalized method?

Author response: Thank you for your comments. Yes, it is a widely used and well-established method of AOAC for mineral estimation. We have followed this method from our published food chemistry article.

Comments: Line 240: The leaves of Lalshak were extracted in 80% acetone to estimate. The extraction method should be succinctly developed.

Author response: Thank you for your comments. For your kind consideration, This is a well-established and widely used method in the literature. We followed the method from our previously published article. To avoid plagiarism, we have written it concisely with citations.

Comments: Line 244: FW!?....for accurate comparison all data should expressed as DW. All data should be recalculated!!!!

Author response: Thank you for your comments. For your kind consideration, some estimation methods of phytochemicals are developed for fresh samples and some are developed for dry samples. For this reason, we have estimated some phytochemicals in fresh weight and some in dry weight.  

Comments: Line 257: idem

Author response: Thank you for your comments. We have revised it.

Comments: Line 259: Fresh leaves were used to determine AsA and DHA. The sample was pre-incubated 259 by Dithiothreitol (DTT)….leaves were ground or not!!! Method should briefly described.

Author response: Thank you for your comments. We have used a widely used and well-established method in our previously published articles like Scientific reports, and food chemistry. In this MS we have written it concisely and briefly with citations to avoid plagiarism. Fresh leaves were thoroughly ground with a a mortar and pestle.

Comments: Line 263: FW. Change to DW and redo calculation

Author response: Thank you for your comments. For your kind consideration, we have estimated this from fresh leaf samples, for this reason, we finally calculated in fresh weight.

Comments: Line 275: Leaves were dried in a shady place to avoid direct sunshine. Did authors checked the DW of this method?

Author response: Thank you for your comments. We have specifically mentioned which one was from dry and fresh samples in the second and third sentences.

Comments: Line 299: the name and nature of the used column should mentioned.

Author response: Thank you for your comments. We have added the name and nature of used column.

Comments: Line 305: according to result section, authors used a MS detector….It is not detailed in this paragraph!!!

Author response: Thank you for your comments. We have added it.

Comments: Line 307: FW: recalculation using DW

Author response: Thank you for your comments. For your kind consideration, we estimated from fresh leaf samples, so we finally calculated as FW.

Comments: Line 323: when necessary, “significant” should be used in all comparison.

Figure 3, 6, 8, 10 and 12, Standard deviation should be used.

Author response: Thank you for your comments. These figures showed the comparison of % of the increase in different salt stresses over the control value/It is the percent increase under different salt-stress conditions over control. So, there should not be any standard deviations and significant test.

Comments: Line 368-374: discussion and explanation about the mechanism response of the plant should be given.

Author response: Thank you for your comments. We have tried our best to explained the response mechanisms.

Comments: Line 425-432: C. spinosum. They stated that….. by 20% and 51%, respectively (Fig. 4). Which physiological/molecular mechanisms are used by the plant?

Author response: Thank you for your comments. We have corrected the figure number. We have tried our best to explained the response mechanisms.

Comments: Line 434: 3.3. Influence of sodium chloride on phytochemicals. The method used for TP determination give Folin–Ciocalteu reducing capacity of sample and not the TP content…..authors should compare their TP content and the content calculated from HPLC data (polyphenols)!...so. Folin–Ciocalteu reducing capacity should be used instead TP content in all sections

Author response: Thank you for your comments. We have changed “TP content” to “Folin–Ciocalteu reducing capacity”. In HPLC analysis, we have determined some flavonoids compounds like flavonols, flavanols, flavones, flavanones, etc. it is not the representation of total polyphenols. Total polyphenols includes simple phenols, phenolic acids, flavonoinds and so on.

Comments: Line 493-500: paragraph discuss clinical results. It is not the subject of this study…. Discussion should focus on response mechanisms

Author response: Thank you for your comments. We have deleted it. We have tried our best to explained the response mechanisms.

Round 2

Reviewer 3 Report (New Reviewer)

dear authors,

In the future, it is preferable to make the calculations at a fixed water content.

Best regards

This manuscript is a resubmission of an earlier submission. The following is a list of the peer review reports and author responses from that submission.

Round 1

Reviewer 1 Report

The first author has published a number of very similar papers with the same model plant species. The only differences are the use of other genotype (which is claimed but not proven even in the original series of papers related to comparison of different genotypes) and some variation in measured parameters. Results itself changes in absolute uniformity due to the effect of the same factors, increasing salinity with no persuasive statistical evaluation. These "results" are accompanied by almost identical lousy conclusions, "supported" by pseudoscientific texts on human health-enhancing effects of various phytochemicals. Most importantly, highly unusual and suspect citation patterns are obvious. Within only two adjacent sentences (lines 56–63), 100 references are introduced in total, from which 6 are self-citations of the first author. From the total  number of references (160, which is far too many for any experimental paper), 47 are obvious self-citations of the first author (28 under the "Sarker U." and 19 under the "Sarkar U.". The most egregious case of inappropriate self-citation is in lines 187–190 of Materials and methods section, where 9 new self-citations are given to support such actions as (•) obtaining averaged replication mean, (•) performing statistical analysis, and (•) application of ANOVA procedure. In addition, the manuscript is full of errors of different categories, both substantive and stylistic.

Reviewer 2 Report

This study lack of innovation and the results demopnstration (figures and table) need to be larlely improved. It lacks attractive to reader and is less improtance to the field of stress study